# Management of Climate Resilience: Exploring the Potential of Digital Twin Technology, 3D City Modelling, and Early Warning Systems

**DOI:** 10.3390/s23052659

**Published:** 2023-02-28

**Authors:** Khurram Riaz, Marion McAfee, Salem S. Gharbia

**Affiliations:** 1Department of Environmental Science, Atlantic Technological University, ATU Sligo, Ash Ln, Ballytivnan, F91 YW50 Sligo, Ireland; 2Centre for Mathematical Modelling and Intelligent Systems for Health and Environment (MISHE), Atlantic Technological University, ATU Sligo, F91 YW50 Sligo, Ireland

**Keywords:** digital twin, early warning system, smart cities, coastal climate change, 3D city modelling, virtual cities, real-time sensors

## Abstract

Cities, and in particular those in coastal low-lying areas, are becoming increasingly susceptible to climate change, the impact of which is worsened by the tendency for population concentration in these areas. Therefore, comprehensive early warning systems are necessary to minimize harm from extreme climate events on communities. Ideally, such a system would allow all stakeholders to acquire accurate up-to-date information and respond effectively. This paper presents a systematic review that highlights the significance, potential, and future directions of 3D city modelling, early warning systems, and digital twins in the creation of technology for building climate resilience through the effective management of smart cities. In total, 68 papers were identified through the PRISMA approach. A total of 37 case studies were included, among which (n = 10) define the framework for a digital twin technology, (n = 14) involve the design of 3D virtual city models, and (n = 13) entail the generation of early warning alerts using the real-time sensor data. This review concludes that the bidirectional flow of data between a digital model and the real physical environment is an emerging concept for enhancing climate resilience. However, the research is primarily in the phase of theoretical concepts and discussion, and numerous research gaps remain regarding the implementation and use of a bidirectional data flow in a true digital twin. Nonetheless, ongoing innovative research projects are exploring the potential of digital twin technology to address the challenges faced by communities in vulnerable areas, which will hopefully lead to practical solutions for enhancing climate resilience in the near future.

## 1. Introduction

The global urban population surpassed the rural population in 2007, thus marking a new ‘urban era’. According to UN projections, 68% of the world’s population is expected to reside in urban areas by 2050 [1]. Moreover, the global population has expanded from 2.5 billion people in the 1950s to 7.8 billion in 2020 [2]. It is anticipated to reach 10.9 billion by the end of the 21st century under the most optimistic scenarios [3].

Coastal areas are particularly populated, with coastal population densities three times the global average [4]. A total of 625 million people are estimated to reside in the low-elevation coastal zone [5]. Coastal communities are highly prone to environmental hazards, including storm surges, coastal flooding and erosion [6,7,8]. In recent years, coastal storms have become more frequent and intense due to climate change [9].

Approximately 38.4% of Europe’s population resides within 50 kilometres of the Mediterranean Sea in Europe’s most densely populated area [10]. Yet, 0.1–1.3 million people in this region are at risk of coastal flooding each year, resulting in losses of about EUR 1 billion in property damage, infrastructure damage, and environmental damage [11]. Significant coastal flooding affected Italy in 2009, Greece in 2013, Spain in 2015/2016, and Portugal in 2020 [12,13,14,15].

To prevent the loss of life and property as a result of catastrophic climatic events in coastal areas, these populated low-lying areas need reliable early warning systems that accurately reflect the region’s complex systems.

In the past, models were used to forecast environmental catastrophes, however, they had limited accuracy due to a lack of data at local scales. The constantly changing parameters of cities make these models unreliable for predicting short- and long-term disasters [16].

Therefore, the notion of real-time data is pursued, with many sensors continuously providing real-time data to the model [17,18], to give a precise depiction of the changing objects and parameters of cities. Real-time data for early warning systems are considered quite reliable. However, a greater problem is the difficulty in the analysis of such data, and how it can be used to give reliable early warnings and provide insights into suitable prevention and mitigation measures in a comprehensible way. Therefore, in recent years, due to advancements in technological, software development, and communication technologies, many cities have started developing virtual city models with real-time sensor data called the ‘digital twin’ [16].

A digital twin (DT) is a digital representation of a physical thing that includes a bidirectional dynamic mapping between the actual object (real time) and its digital model. The University of Michigan pioneered the concept of the digital twin in 2002. NASA introduced the digital twin concept to develop physical models and simulations of spacecraft for the first time in 2010. Later, digital twins were employed in the construction of buildings, energy flow, and improving product manufacturing in industries [19,20]. Moreover, a digital twin must be linked with a real-world item; otherwise, it is only a model [21].

Digital twins involve bidirectional mapping between the real world and the virtual world. This is different from a unidirectional mapping that only translates data from physical items to digital objects, a concept known as a digital shadow [22,23,24]. Digital twins, on the other hand, have the ability to enable virtual entities to automatically influence physical objects, as shown in Figure 1.

According to [24], developing digital twins in the context of smart cities has evolved from the initial level of static 3D modelling to the level of the digital twin, which combines dynamic digital technology with a static 3D model. At a fundamental level, the word “smart city” has no recognised definition, and it was first used in the 1990s [25]. Smart cities are urban centres that focus on a secure and safe environment and efficient utilities combined with digital technology [26].

According to [27], advances in technology, particularly geomatics, information technology, and GIS, have renewed interest in the concept of digital twins in smart cities. The real-time resilience of city infrastructure can be assessed through sensors, allowing experts to evaluate performance during catastrophic events like climate change. Linking all aspects of a city to a digital twin in the cloud makes it easier to monitor performance and detect flaws [23,26,28,29,30]. Wireless, IoT, and 5G sensor technologies enable cities to collect up-to-date, site-specific data in real-time, generated by sensors and cameras distributed throughout the city. This helps urban designers and stakeholders to work more effectively on approaches to manage cities, as illustrated in Figure 2.

In the context of climate change and especially coastal adaptation, sensors may gather data regarding various weather variables in order to monitor coastal hazards. Once processed with different algorithms and statistical tools (machine learning [31], big data [32], cloud computing [33,34,35], and artificial intelligence [22,36]), using various geo navigation and referencing platforms such as WGIS and ArcGIS, the data can be monitored and visualised for information exchange between city decision makers and stakeholders in order to implement mitigation measures [27].

The frequent sharing of data between digital and physical twins throughout their shared lifecycles may enable smart cities to learn from the insights and adapt over time [37]. This would allow the city to anticipate and react to disasters like floods, erosion, sea-level rise, and even weather forecasts much more rapidly and effectively. In order to prevent climate change and boost early warning signals, urban planners in Helsinki, Finland and Singapore have already started creating a digital twin for the climate atlas [38]. Similarly, Las Vegas and New York City have already begun to develop digital twins for use cases such as developing net-zero carbon emission buildings to increase sustainability and save energy [39]. The top smart city adopters are Singapore, Zurich, and Oslo, as per the smart city index. Singapore has invested $73 million in its Virtual Singapore project, allowing for experimentation with wireless networks, identification of solar panel locations, and climate resilience enhancement [40].

In recent years, the development of smart cities has become increasingly important due to the need for increased climate resilience. In this systematic review, we explore the three main categories related to digital twins in the context of the climate resilience of cities, namely virtual 3D models, early warning systems, and digital twins. We describe the use of 3D modelling for natural disaster management, real-time sensor data for generating early warning systems, and the merging of models and sensor data to create digital twins. To provide a comprehensive review of these categories, we analysed 68 articles obtained from three different databases and summarised the tools, software, applications, and sensors used by the authors in their studies. By linking these categories and explaining how 3D modelling and real-time sensor data are integrated for creating digital twin technology, this review sheds light on the future research potential and barriers in this field.

Moreover, we outline some ongoing projects from across the world that are working to develop digital twin technology for enhancing climate resilience in smart cities. This study briefly outlines most of the typical techniques for developing a digital twin for producing an early warning system for extreme environmental hazardous events, as described by a number of researchers. We envisage this report will be extremely valuable for future researchers and practitioners who are trying to create smart, climate-resilient cities utilizing digital twin technologies.

## 2. Materials and Methods

The purpose of this paper was to identify the state of the art of the growing digital twin technology used for smart city climate resilience via an early warning system. We followed the preferred reporting items for systematic reviews and meta-analyses (PRISMA) principles [41,42,43]. The PRISMA flow chart depicts the whole process of finding, identifying, screening, verifying, and summarising the material for this systematic literature review. PRISMA includes a 27-item checklist and a flow chart that assists systematic reviewers in correctly, concisely, and thoroughly identifying eligible papers [43].

### 2.1. Data Identification

Finding relevant past published articles for this paper was accomplished via the use of the keywords digital twin, climate change, smart cities, climate resilience, city modelling, and GIS-based. These were combined as shown in Table 1 using several scientific databases, viz, Web of Science, ScienceDirect, and IEEE Xplore. The primary reason for using these keywords is that the concept of a digital twin is a broad topic that has been extensively researched in the medical, manufacturing, and production fields. We utilized terms such as climate, climate change, and climate resilience in order to narrow the selection of publications to those relating to climate change using digital twin and real-time sensor data. Additionally, significant inclusion and exclusion criteria were applied throughout all the databases with different keywords as depicted in Figure 3 and Table 1. Figure 4 illustrates the outcomes of the systematic analysis of the reviewed articles conducted with VOS viewers through the co-occurrence method.

### 2.2. Data Screening

According to the search keywords and selection criteria, a total of 1601 articles were extracted from all three databases, with 817 papers (51.03%) collected from the ScienceDirect database, and 650 papers (40.59%) extracted from the Web of Science database. Similarly, a total of 134 (8.36%) publications from the IEEE database were retrieved between 21 November 2022 and to 21 December 2021. The initial exclusion criteria were applied to eliminate duplicates and covered the period from 2010 to 2021, resulting in the exclusion of about 691 articles. After the first screening, 910 articles were selected for the title and abstract screening, which is the second screening test for the eligibility of the full-text article. Approximately 733 papers were excluded in the second screening using different excluding keywords (production flow, 80 papers, 7.46%; building technology, 119 papers, 11.11%; construction, 70 papers, 6.5%; transport, 31 papers, 2.89%; health care, 44 papers, 4.1%; economy, 35, papers, 3.26%; blockchain, 48 papers, 4.48%). The third stage was the final ‘results’ section, where the papers undergo a full text review. Then, 68 papers were selected for the study. The review of these papers is divided into three main sections. These are 3D city models, real-time sensor data for early warning alerts, and digital twin technology.

## 3. Results

The first stage in a systematic review is categorising the final selected articles from the PRISMA screening. As described in the methodology section, a total of 68 articles were selected. Four major types of articles were identified: case studies, general studies, review papers, and conference papers, with a total of 37 case studies, 15 general studies, 5 review papers, and 11 conference papers selected. Figure 5 highlights the number of case study publications by country. Most case studies were conducted in China with seven case studies. Four were in the United States, three in each of Spain and Taiwan, and two each in Italy, Singapore, Bangladesh, Norway, and Korea, Other single case studies were conducted in France, Ireland, the UK, Germany, India, Russia, Japan, Saudi Arabia, Oman, Finland, Malaysia, Greece, and New Zealand.

Figure 6 summarises the article’s broad thematic categories, highlighting the various elements which are brought together to create a digital twin for the purposes of climate event management. Therefore, it is important to study the previously developed technologies used for the identification and early warning of extreme climate events. Moreover, as discussed in the introduction, a digital twin is basically a concept of bringing real-time data from the physical world to the virtual world and based on analysis in the virtual domain, automatically implementing some change to the physical world. Therefore, the aim of this categorisation is to help the reader better grasp the study concept, interlinkages between these concepts, and their future direction.

The following sections provide a brief summary of the technologies required for the creation of a digital twin, building on the works of numerous scholars who have previously used these tools in the development of city modelling and early warning alert systems. This review provides a summary and discussion for comprehending the numerous techniques, methodologies, advantages, and limitations in the research conducted to date towards the creation of digital twins for the management of extreme climate events. We anticipate that this study will be highly valuable to researchers and practitioners who are working on integrating 3D city models with real-time sensor data in order to provide early warnings of climatic extremes and determine optimum mitigation and response strategies.

### 3.1. Category: Virtual City Modelling

With rising populations and urbanisation, it is becoming more challenging to manage the impacts on cities from different environmental catastrophic events [44,45,46]. As a result, the concept of city modelling has emerged for the detailed management of complex urban cities. City models are characterized and represented by two-dimensional (2D), three-dimensional spatial (3D) data, and georeferenced data. A three-dimensional (3D) city model is a computerised representation of a city’s topography, sites, structures, vegetation, infrastructure, and landscape components, as well as associated items that belong to urban regions. One of the key characteristics of creating a city model is the geomatic technique.

The main geomatic techniques for creating 3D city models are photogrammetry, geographical information system (GIS), remote sensing (RS), a global positioning system (GPS), and radar-based and laser-based systems. Among these techniques, the most common technique is GIS-based city modelling. Some of the recent publications on GIS-based 3D city modelling are [47,48,49,50,51]. Another very popular GIS technique used for creating city models by various researchers is geodesign. Geodesign refers to a process that involves the integration of geographical information with design thinking to support decision-making for a wide range of environmental and social issues [52]. For example, reference [46] used a geodesign technique to identify gaps in the transition of an urban lagoon toward flood resistance. Their research indicates that the study area is very vulnerable to flooding during periods of severe rainfall, with subterranean floors and buildings being significantly more prone to floods.

Another emerging technique for creating 3D city models is geospatial technology. Geospatial technology is the combination of GPS, GIS, and RS. It facilitates the collection of georeferenced data for use in analysis, modelling, simulations, and visualization. For example, Ref. [53] proposed the use of geospatial data for creating a 3D city model with various levels of details gathered by laser scanning photogrammetry. Their results show that this method can capture morphological change in the cities such as buildings, trees, and open areas. Moreover, they also conducted a case study in the city of Prerov, Slovakia, which experienced a rapid transformation from an agricultural and seminatural landscape to the largest residential area in the city in the 1960s.

CityGML is another well-known 3D city modelling platform. City GML is an acronym for city geography markup language. It is a GML application schema and information model for exchanging 3D city and landscape models. CityGML is a generic information model for expressing geovirtual 3D environments, such as 3D representations of virtual cities. It presents classes and relationships for topographic elements in urban and regional models. As seen in Table 2, a significant amount of environmental research involving 3D city modelling uses CityGML.

These models are created with different levels of details (LOD) to provide multiple resolutions and different levels of abstraction. Other metrics, such as the level of spatio-semantic coherence and resolution of the texture, can be considered a part of the LOD. For example, CityGML defines five LODs for building models.

LOD 0—Regional, landscape;LOD 1—City, region;LOD 2—City districts, projects;LOD 3—Architectural models (exterior, landmarks);LOD 4—Architectural models (building interiors).

Table 2 summarises the research works identified in this systematic review pertaining to 3D city modelling, where the LOD associated with each research paper is specified. For example, Refs. [54,55] provided a comprehensive risk assessment tool to aid in the development of mitigation strategies for severe precipitation, sea-level rise, and storm surge in Spain’s urban areas. They use the CityGML open urban-data modelling tool in both studies with LOD1 and LOD2 details. The main objective of their study was to assist evidence-based decision-making with respect to mitigation measures via an unorganised and objective method of data analysis supported by a graphical 3D display. Similarly, numerous researchers used the same 3D modelling technology (CityGML) to allocate resources and design plans for a variety of climate-related hazards. See [44,47,51,56,57,58] for some recent publications utilising CityGML modelling tools.

CityJSON, which implements the data model of CityGML using JavaScript object notation (JSON) encoding, has emerged as one of the advances in 3D city model formats. The authors contend that one of the advantages of CityJSON encoding is that it may be easily read in online systems (such as web browsers) that support JSON encoding [59,60]. In addition, the majority of computer languages may integrate the necessary structures by combining two fundamental data structures, namely ordered lists and key-value pairs [60]. Due to the fact that CityJSON is still a relatively recent product, its usage has not yet been thoroughly covered in the literature.

**Table 2 sensors-23-02659-t002:** List of eligible virtual 3D city model articles included in the review with their research objectives and other contributions.

Citation	Region	Application	LOD	3D Modelling	Case Study	City Model	Real-Time Data	DT	Tools Utilised	Software Utilised
[45]	Taiwan	Sea level rise	LOD1	3D	√	√	×	×	Block model + generic texture model + photo-realistic economic model + photo-realistic detailed model.	Inet series + ArcGIS, +TerraSuite, +GeoBeans3D
[53]	Slovakia	Urban land management	LOD3	3D	√	√	×	×	Laser scanning + photogrammetry + electronic tachymetry + differential positioning through satellites	SketchUP + CAD + GIS (ArcGIS, GRASS GIS)
[46]	Taiwan	Flood	N/A	3D	√	√	×	×	Integrating GIS with Geodesign + rule-based modelling	ArcGIS Pro 2.0 + CityEngine 2017.0
[61]	Italy	Flood	LOD1	3D	√	√	×	×	LIDAR data + Texture mapping	ArcGIS^®^ 10.3 + SketchUP
[44]	Korea	Flood	LOD1	3D	√	√	×	×	City Geography Markup Language (CityGML)	ArcMap + Hancel (Hancom Office)
[62]	China	Flood	Not mentioned	3D	√	×	×	×	LiDAR + Aerial imagery (UAV) + texture mapping	SketchUP
[57]	Germany	Flood	LOD1 + LOD2	3D	√	×	×	×	CityGML + virtualcityMAP	SQL queries + GIS software
[56]	Italy	Flood	LOD2	2D + 3D	√	√	×	×	3D point clouding + Terrestrial laser scanning + Airborne laser scanning + LiDAR + CityGML + Texture mapping	SketchUp + ArcGIS^®^ 10.3 + CloudCompare
[54]	Spain	Flood	Not mentioned	3D	√	√	×	×	LIDAR + CityGML	Not mentioned
[58]	China	Flood (waterlogging cities)	LOD1	3D	√	×	×	×	CityGML	JDK, PostSQL, Tomcat
[51]	General	Flood	LOD1	N/A	×	×	×	×	CityGML + Web 3D GIS + X3D + WebGL + X3DOM + GeoServer3D + jQuery	Linux (Ubuntu) + Blender + SketchUp
[47]	Oman	Flash flood	LoD1	3D	√	√	×	×	CityGML + Watershed Modelling System + Visualising tools (FZK viewer and Cesium)	ArcGIS + PostgreSQL-PostGIS + FME engine
[55]	Spain	Flood	LOD1 and LOD2	3D	√	√	×	×	LIDAR + DTM + CityGML	Not mentioned
[63]	Spain	Decarbonisation	N/A	N/A	√	×	×	×	ENER-BI + Cities4ZERO	QGIS + PowerBI + Influxdb, PostgreSQL + Postgis + EnergyPlus + EnergyPLAN
[64]	General	Urban microclimate	N/A	N/A	×	×	√	√	Satellite sensors	Not mentioned

The primary data input techniques for the aforementioned models are aerial image data, satellite image-based data, terrestrial close-range image data, and a hybrid approach (a combination of the above methods).

Airborne data is the most common method for the collection of 3D city model data. An example of remotely sensed data using drones is light detection and ranging (LIDAR) [54,61,62]. Two studies were reported on the visualisation of flood events by the same authors in the town of Cosenza, Calabria, Italy using LIDAR data [56,61].

Laser scanning involves the controlled deflection of laser beams, visible or invisible, used for capturing the 3D geometry of buildings, objects, sites, and infrastructure. For example, Ref. [56] describes the possibilities and perspectives of employing terrestrial and aerial laser scanning to generate 3D flood hazard maps in urban areas. In addition, they conclude that a point cloud source is now the best option for hazard mapping. A point cloud is a collection of points of data in space. Every point has its own set of coordinates. These point clouds are often generated by the above-mentioned tools (airborne, terrestrial, and hybrid methods) that measure a large number of points on the exterior surfaces of objects. Point clouds are increasingly being used in creating 3D city modelling as this facilitates the definition of buildings and other structures as a collection of points with their respective coordinates. This has benefits, e.g., the evaluation of flood hazards in the city. After an appropriate mesh is constructed based on the combination of airborne and terrestrial-based laser data. The visualisation of flood levels may be evaluated, without the need to build a legend, by simply clicking directly on the location of interest, which contains the topographic coordinates as the attributes. Using this technology, it is feasible to read the water level reached at any point on the vertical surfaces of the at-risk objects in the virtual pictures of the flood [56].

Table 2 lists the primary software utilized for simulation, integration, and visualization of real-time data coming from different sensors, such as SketchUp, ArcGIS CityEngine, G2 Deals, ArchiCAD, CityCAD, Giraffe, and TestFit. SketchUp is regarded as the most common software application for creating 3D city models [65]. It is a computer tool for 3D modelling used in a variety of drawing and design applications. Numerous authors have conducted case studies using SketchUp, some of which include [51,56,61,62,66]. ArcGIS is the other primary software used in the studies captured in this review [44,45,46,56,57,61,63,67]. ArcGIS is recognized for producing 3D city mapping and geodesigns from aerial image-based data, satellite image-based data, and terrestrial close-range image data.

It is important to consider how the data, tools, and software used for 3D city modelling are connected to the concept of digital twin technology. As stated in the introductory paragraph, the digital twin of a smart city is essentially a digital representation of the physical world, in which the data is continually transmitted between this virtual system and the real world. Because of this, an accurate virtual model of the city is needed in order to capture and process the real-time data in the digital twin. As a result, we present this summary of the methodologies and tools that are currently being used in research on the digital representation of cities.

### 3.2. Category: Early Warning System

The term ‘early warning’ is used in a variety of sectors to refer to the giving of information about an approaching hazardous situation in order to allow action in advance to mitigate the hazards. Natural geophysical and biological hazards, complicated socio-political events, industrial hazards, human health concerns, and a variety of other threats all have early warning systems. However, in this systematic analysis category, we focus on geophysical hazards only, such as flooding, storm surge, heavy storms, erosion, land sliding, droughts, and tsunamis, as well as associated coastal hazards.

The number of early warning system papers included in this systematic review is shown in Table 3, along with a high-level summary of their attributes. To create an early warning system capable of generating alerts, it is necessary to make information accessible to stakeholders and to identify an impending risk in time. In most situations, stakeholders can take corrective measures, though often the window for preventative action has gone. This is where real-time data comes into play. When seen as an early warning signal, real-time data enables stakeholders to effectively take precautionary action.

There are many publications on early warning systems in which real-time data is used to monitor and generate warnings for geophysical hazards, some examples are [17,68,69,70,71,72,73,74,75,76,77,78,79].

The basic fundamental element comprising the overall structure of end-to-end early warning alerts for hazard forecasting is data collection in real-time for the purpose of predicting the severity of a hazard, including the moment at which it will begin and the scope and scale of it. The data acquisition typically consists of a field unit and a remote network interface. The field system can include different sensors such as rainfall, temperature, humidity, soil temperature, soil moisture, tiltmeter, and so on, installed with solar panels for powering the field unit. For example, Refs. [72,74,77] use water level sensors; Refs. [71,80] use tide gauge sensors; Refs. [81,82,83] rain gauge sensors; and [79,81] use temperature and humidity sensors, for monitoring the landslide and flood early warning systems.

Increasingly, researchers are using micro-electro-mechanical systems (MEMS) sensors for early warning systems for landslide and flood occurrences, as well as earthquake magnitudes. MEMS technology relates to the creation of tiny integrated devices or systems that combine mechanical and electrical components [84]. The primary reasons for the increasing interest in MEMs technologies are the low cost and small size. Moreover, wireless sensors based on MEMS are rapidly rising in popularity for real-time monitoring of geological hazard occurrences such as landslides [78]. Similar research was conducted in the Himalayan area of India by [72] to provide an early warning system for landslide occurrences. The study utilised wireless sensor networks based on MEMS containing a tiltmeter sensor with a control circuit to monitor the level of the soil.

Laser and nonlaser crack sensors have also been widely used by researchers over the last decade to monitor landslide occurrences in real time. The authors of [70] presented a report in 2011 for real-time landslide event data monitoring utilising force and crack sensor data.

The authors of [18] propose a knowledge management system for flood decision support in Jakarta that collects data from various sources and provides a dashboard for decision-making. It has the potential to enhance existing flood management efforts, though challenges related to data management and governance need to be addressed. The system could be extended with additional applications such as real-time decision making, data and analytical management, and asset management. The authors of [74] undertook research to track flood events in the United States of America state of Virginia. Ultrasonic sensors and water level monitors were used to collect real-time data on floods. These sensors improved on previous prototypes to sense water levels inside a drainage system in order to offer a more accurate image of overloaded or failing infrastructure. Similarly [85] also used ultrasonic sensors to develop a real-time flood monitoring and early warning system in the northern portion of Isabela. Their research is limited to water level detection and an early warning system (website and/or SMS) that warns concerned authorities and citizens of a possible flood occurrence.

In addition to producing flood and landslide early warning systems, there are a number of additional natural hazards which are also the target of early warning system research. For example, after the 2004 Indian Ocean mega tsunami, a tsunami early warning system and other tsunami mitigation initiatives were implemented in the European and Mediterranean regions. GITEC (1992–1995) and GITEC-TWO (1996–1998), which were supported by the European Union (EU), highlighted the need for such activities long before 2004 [80]. The authors of [71] published a recent paper on monitoring and providing an early warning alert system for tsunami events through inundation predictions within a short period of time. Inundation is one of the direct causes of devastation and damage caused by tsunamis. The inundation data utilised in this research is derived from the TsunAWI tsunami modelling system. The authors used a data analysis process to generate an early warning system for tsunami hazards. The data analysis process consists of data collection, data transformation, data analysis (using GIS analysis, predictive analysis, and basic statistical analysis), and data integration. As a result, the tsunami impact prediction system produces estimates of the depth and distance to which waves will reach individual cities in an emergency situation, based on the integrated tsunami intensity scale (ITIS-2012). In addition, the system automatically detects sea level anomalies using tide gauge sensors and a tsunami detection algorithm.

The next step is determining how to use the sensor data in order to set the threshold for hazard warnings. For that, the data received from all of these sensors are stored in either a local or cloud-based data logger. The traditional way to store the data is through a local data logger via a micro-SD card. These SD cards are installed with a global system for mobile communication module (GSM), which allows the data to be locally stored and at the same time transmit the data to a remote server via a sim card on a 4G or 5G network.

**Table 3 sensors-23-02659-t003:** List of eligible early warning system articles included in the review with their research objectives and other contributions.

Citation	Region	Category	Application	IoT	Case Study	Early Warning System	Real-Time Data	DT	Sensors Mentioned	Software Mentioned
[74]	US	Conference paper	Flood	√	√	×	√	×	Ultrasonic sensor + Water level sensors	Sqlite3 + AWSEC2
[86]	General	Review paper	Landslide	×	×	√	√	×	Wireless network sensors	Kubernetes + Docker
[87]	General	Review paper	General Study	√	×	×	×	×	None	None
[88]	US	Book	Flood	×	×	√	√	×	NEXRAD radar + Telemetric tipping bucket rain gauge	xgboost
[89]	Bangladesh	Case study	Landslide	×	√	√	×	×	None	Rasterpack Python + mysql + rclimdex + GDAL
[68]	Malaysia	Case study	Landslide	√	√	√	√	×	Satellite and airborne monitoring	Arcsde v9.3 + Geomatica 10.1 + arcgis + JAVA Enterprise Edition (J2EETM)
[75]	Taiwan	Case study	Landslide	×	√	√	√	×	Not mentioned	QPESUMS + rilews
[90]	Europe	Review Paper	Flood	×	×	√	×	×	Not mentioned	Not Mentioned
[81]	India	Case study	Landslide	×	√	√	√	×	Rainfall Sensor, Temperature, Humidity, Wind speed, Wind direction, Soil moisture, temperature and Tiltmeter sensor.	Not Mentioned
[70]	China	Conference Paper	Landslide	×	√	√	√	×	Force sensor, laser sensor, crack deformation sensor, data Acquisition and transmission device, power supply device, and communication antenna	Not Mention
[80]	Europe	Book	Tsunami	×	×	√	√	×	Seismic alerting devices (consist of eight sensors) + Radar-type (ultrasonic) tide gauges	Not mentioned
[91]	Cyprus	Case study	Flood	×	√	√	×	×	None	Decatastrophize (DECAT)
[76]	General	General Study	Disaster management	√	×	√	√	√	General	General
[72]	India	Case study	Landslide	×	√	√	√	×	Wireless Sensor Networks based on microelectronicmechanical systems + Tilt sensor + Water content sensors	Not mentioned
[73]	China	Case study	Flooding	√	√	√	√	×	Remote sensing stations + RFID + mobile phones + webcam + Other sensors	Not mentioned
[71]	Indonesia	Case study	Tsunami	×	√	√	√	×	Tide gauge sensors	GIS software + QGIS
[79]	China	Conference paper	Landslide	×	√	√	√	×	Temperature and humidity sensors + HD cameras + Panoramic camera	Arcgis + arcxml + javascript + Java API + arcsde
[78]	General	Conference paper	Landslide	×	×	√	√	×	Wireless Sensor Networks based on microelectronic mechanical systems	Arcgis + webgis (arcims) + Temporal GIS + arcsde
[77]	Greece	Conference paper	Flood	×	×	√	√	×	Water Level Sensor + SYMMETRON’s Stylitis 20	Webgis (mysql) + Node.js + Dygraphs JS + sdks
[69]	General	Conference paper	Flood	×	×	√	√	×	General	General
[82]	Myanmar	Conference paper	Landside	√	√	√	√	×	Rain gauge sensor + Accelerometer sensor + Soil moisture sensor + Temperature sensor + Piezometer sensor	Not mentioned

One of the most recent papers published by [81] deals with the design and implementation of an early warning system that uses low-cost sensors to provide real-time landslide data over the Himalayan region of India. They used sensors including rainfall, temperature, humidity, soil temperature, soil moisture, and tiltmeter for monitoring landslide movements, with an independent power source of a 50 W solar panel. Using the GSM installed at the field unit, they could continually acquire real-time data, compare it with threshold limits, and transmit emergency warnings in the form of different levels (orange, red, and green alerts). As an outcome of their results, they conclude that this method may also be used to calibrate mainstream flood forecasts.

Alternatively, cloud storage is a concept of cloud computing that enables significant real-time data storage in remote third-party servers. Internet users may access these servers at any time. Cloud storage is also known as utility storage. Several recent works have exploited cloud storage for data logging and networking in early warning systems [73,76,79,86,87,90].

A platform currently being used by several researchers is the webGIS platform. WebGIS facilitates client–server information exchange through the web. WebGIS may be accessed from anywhere using a web browser, and its databases are stored in the cloud. The study in [77] employed a water level sensor to monitor the floods of the Arachthos River. They share their findings on a web-based GIS platform called WatchArachthos, which gives local authorities the ability to monitor the river and its current status, enhancing flood crisis management effectiveness and efficiency. Some other examples of the articles that have been published by the various authors using the Web-based GIS platform include [76,77,78,79,83,89,90,91,92].

The final phase in the design of an early warning system for natural hazards is the use of predictive analysis and basic statistical analysis applied to the sensor data in order to determine the threshold limits for the relevant hazard. Then, those predictions are compared to the threshold limit for that hazard in a certain area or city. The alerts are usually characterised into three main colours categorised as red (high alert), orange (medium alert), and green (low alert). Previously, the early warning system for a particular hazard in a specific region was created in a unidirectional manner. Experts generated the warnings by transmitting sensor data in real time to the system and using various algorithms to generate the alerts. However, due to technological advances in AI and ML, experts are exploring digital twin technology for early warning systems [31].

The main advancement between traditional unidirectional early warning alerts and the bidirectional (digital twin) early warnings system is the use of 3D and 4D city geometry data. Another significant advancement is the concept of using data analytics and machine learning to build simulation models that can be updated and changed in real-time as their physical equivalents change and vice versa. The following section will elaborate on how these traditional unidirectional early warning alerts are beginning to be implemented in digital twin technology by the automatic bidirectional flow of data.

### 3.3. Category: Digital Twin

The authors of [93] define the digital twin as an idea that enables the creation of virtual representations of physical items by receiving sufficient data gathered remotely through sensor networks from the internet of things (IoT), i.e., connected appliances, smart factory equipment, wearable health monitors, smart home security systems, and smart cars. In the context of cities, digital twins integrate real-time remote monitoring for more effective decision making in the event of a disaster [40]. Emerging digital technologies have the potential to offer more effective, quick, and reliable resilience assessments and allow improved decision-making based on actionable performance indicators prior to, during, and after the occurrence of hazards.

A digital twin can aid in the cocreation and testing of scenarios involving a particular town or shoreline parameter. For example, if a flood event occurred, how much damage may occur to that particular town? [94]. The digital twin requires a substantial amount of observation to predict future situations and evaluate, for example, nature-based solutions for such catastrophic events.

In this subsection, we concentrate on the studies that have been conducted to determine the environmental impacts of catastrophic natural disasters such as floods, landslides, and sea-level rise utilising digital twin technology. Table 4 depicts the number of digital twin papers included in this systematic review along with their associated details.

Flooding is one of the most prevalent natural hazards on a worldwide scale, inflicting more damage than any other severe weather phenomenon. According to the 2020 global natural disaster assessment report, floods were the most frequent of all types of natural disasters around the world in 2020. In total, 193 floods affected 201 countries [95]. As cities’ sometimes insufficient drainage infrastructure is pressurised as a result of flood events, the lives and livelihoods of citizens became more vulnerable [96]. The European Commission launched an initiative in 2022 called destination earth (destinEarth) to establish a digital twin for urban flood simulation in order to increase city flood resilience via data-driven planning, development, and operation. This initiative will get an initial EUR 150 million in funding, to develop a complete digital reproduction of the Earth by 2030 [28].

Combining the data on the 3D city geometry with data from real-time sensors and employing a variety of statistical algorithms in order to create an early warning support system are the fundamental elements that make up the overall structure of the digital twin technology. The aim is to generate hazard forecasts and the creation of cities that are resilient to the effects of climate change. There are many different technologies and methods being used by different researchers for obtaining relevant real-time data. Several sensors and data collection methods were outlined in Section 3.2 for flood, landslide, tsunami, sea level rise, temperature, and water level monitoring. Similarly, different approaches for collecting the city geometry data such as aerial images, satellite images, terrestrial close-range images, and hybrid approaches were discussed in Section 3.1. However, the infrastructure of real-time sensors is likely insufficient at present to provide dynamic spatiotemporal information about the physical vulnerabilities of complex cities. Moreover, due to continuous changes in the city geometry, it is insufficient to completely rely on the traditional methods of capturing information. Therefore, there are some new approaches through which data is collected by engaging a citizen science approach such as crowd-sourcing data, user tagging simulation data, smartphone apps, social sensing, social media messages and Skyline simulation.

Crowdsourcing data is now a well-known method for capturing real-time data. Crowdsourced data collection enables researchers to outsource basic tasks or surveys at a low cost, collect data in real-time, and gain considerably more observations than with conventional data collection, despite its comparatively low cost. For example, Ref. [97] reported a case study in Dublin, Ireland utilising a 3D virtual representation of a smart city using real-time data from various sensors. They utilised crowd simulation information with multimodal sensing data and an interactive computer-aided virtual 3D city model. The idea was to assist stakeholders and urban authorities in evacuating flood-affected people. Similarly, a case study in Houston, Texas, was completed with the use of crowdsourced information, multimodal sensing data, and an interactive computer-aided virtual environment to assess natural hazard risks such as flood and inundation [98]. The authors of [99] conducted research on urban flood monitoring with photos collected from a crowdsourcing app along with Twitter messages.

Social sensing generally refers to a group of sensing and data-gathering paradigms in which information is gathered from individuals or devices on their behalf. Some well-known examples are tweets, locations apps, Facebook, Flickr, and other smartphone apps (e.g., rapid inundation mapping from photos posted on Twitter [100], rainfall-runoff estimation and flood forecasting with social media messages [101], generated real-time data of flooding through social sensing (tweeted pictures) [102], near-real-time flood map based on social media messages [103], and rapid inundation mapping from photos posted in Twitter [100]).

In the skyline simulation, a three-dimensional model is made accessible to the public for editing and enables the simple removal and installation of newly proposed buildings or structures. Using building information modelling (BIM), any suggested building designs may be readily integrated into the digital twin. BIM is a process that involves the development and maintenance of digital representations of the physical and functional properties of places [104]. Citizens and public authorities would then be able to stroll around the digital twin and witness the impact of the new structure on the skyline from different points. With an exact digital twin accessible, users might flag real-world issues in the model and have the information sent to the appropriate government agencies.

**Table 4 sensors-23-02659-t004:** List of eligible digital twin articles included in the review with their research objectives and other contributions.

Citation	Study Type	Region	Case Study	Real-Time Data	3D Model	Applications	Tools Utilized/Describe
[105]	Journal Article	New Zealand	√	√	√	Wetland (Flood and storms)	IoT sensors + Social sensors data + thingworx + AR UX
[97]	Journal article	Ireland	×	×	×	Flood	Skyline simulation + flood simulation + crowd simulation + User tagging simulation
[106]	Review Paper	General	×	×	×	Climate resilience	N/A
[107]	Review Paper	Saudi Arabia	×	×	×	Flood	N/A
[16]	Conference Paper	General	×	×	×	General	N/A
[108]	Journal Article	China	√	×	√	Flood	3D hydrodynamic mode + Digital aerial photogrammetry + Oblique images +
[109]	Review paper	General	×	×	×	Smart city management	CityGML+ 3D city model
[110]	Journal Article	Singapore	√	√	√	Urban landscape management	LiDAR + Point cloud modelling + geo-specific digital 3D Models (Laser scanning) + Voxel model + Computational fluid dynamics
[111]	Journal article	Norway	√	×	√	Extreme climate modelling	FKB-Laser10 + Digital elevation maps + GIS-tool Calamity Levels of Urban Drainage Systems
[112]	Review Paper	General	×	×	×	Smart City management	N/A
[113]	Journal article	China	√	×	√	Smart city management (Heat Island/ Forest cover)	Remote sensing + satellite image data
[114]	Journal Article	South Korea	√	×	√	Smart city management (carbon emissions)	Raw Data + 3D building data + the carbon emissions values of the four factors estimated including the creation of a city model (LOD 1)
[115]	Journal Article	US	√	√	√	Smart city management (carbon emissions)	CityGML + OGC Sensor Things API standard
[116]	Review Paper	General study	×	×	×	Sustainability	N/A
[117]	Journal Article	Norway	√	√	√	City Management	Traffic visualization + Heat maps + 3D city models
[118]	Conference Paper	General	×	×	×	Smart city management	N/A
[119]	Conference Paper	Japan	×	×	×	Smart city management	N/A
[120]	Conference Paper	Finland	×	×	×	Smart city management	CityGML Explanation
[37]	Review Paper	General	×	×	×	Smart City Management	N/A
[93]	Conference Paper	General	×	×	×	City Management	Surveillance cameras + pressure sensors + humidity sensors + temperature sensors + air pollution sensors + geolocation data of vehicles + information from passengers’ travel cards
[121]	Journal Articles	Germany	√	√	√	City management	3D Street network model with space syntax + Urban mobility simulation with (SUMO) Simulation of Urban Mobility + Airflow simulation + Sensor network data (Temp, humidity), + social data
[102]	Journal Articles	General	×	×	×	Flood	Social Sensing (graph-based approach + image-ranking algorithm + Situational Awareness)
[94]	Journal Articles	General	×	×	×	Disaster city management	N/A
[98]	Journal Articles	US	√	√	√	Disaster city management	Participatory sensing + crowdsourced visual data
[122]	Review Paper	General	×	×	×	General Study	N/A
[123]	Journal Articles	General	×	×	×	Agricultural sector	WebGIS + Satellite imagery + remotely sensed data
[124]	Journal Articles	General	×	×	×	Disaster city management	Remote sensing + Synthetic aperture radar + LiDAR + Optical and thermal range + social sensing + crowdsourcing

Recent research has demonstrated the potential of digital twin technology for various applications in urban planning and environmental management. For instance, in a study of Dublin, Ireland, researchers [97] used skyline simulation to enable the public to investigate suggested changes to urban policy and planning through an online 3D model of the city. The proposed modifications were made freely accessible and visible to the public before implementation.

Another work [105] describes a case study addressing the real-world challenge of maintaining a large number of wetlands in a large area with significant value, including real-time monitoring, remote control, prioritisation, prediction, immediate alerting, and scheduling maintenance functionalities. Additionally, his research proposed the concept of ‘Digital Twin as a Service’ to manage the maintenance plan for over 500 distinct wetlands in a 2400 square kilometre density zone in New Zealand. Digital twin as a service is proposed as an ongoing tool to manage each wetland, and monitor, regulate, and eventually forecast real-world consequences during routine maintenance.

In addition to climate resilience, digital twin technology has been applied to various other fields, including urban planning and management. For example, Ref. [125] proposed a digital twin framework for the discharge of the France Calais Canal, which would use real-time data to forecast future scenarios and improve the management of hydrological systems.

In a study on digital twins and 3D city models in Saudi Arabia, Ref. [107] emphasised the importance of GeoICT in the creation of smart cities. GeoICT is a beneficial technology for cities to evaluate urban sustainability and the policy implications that go along with it, particularly when it comes to flood assessment and monitoring in the country. Navarra, 2013 defines GeoICT as the concept of integrating geospatial technology into the mainstream of ICT. Another study [108], created a 3D hydrodynamic model using digital aerial photogrammetry and BIM and GIS technology. This 3D model shows that numerical hydrology and flood simulations based on digital city models may benefit from more precise flow information, such as flood extent, axial velocity, and vortex structures, as shown by the comparison of findings between 2D and 3D approaches. Moreover, both the DHI MIKE 21 and the Flow-3D software packages were used in the study to simulate coastal flood inundation in 2D and 3D city models. However, the authors concluded that for more accurate results in the future, real-time sensor data feeds should be utilised in future studies.

One potential avenue for improving disaster response and emergency management is through the use of digital twin technology, according to a proposal by Fan et al. [102]. They propose that the field of information technology and artificial intelligence may come together in the context of disaster response and emergency management using a digital twin paradigm. Using a multifactor game theory approach, the suggested paradigm consists of four parts: multidata sensing for data collection, data integration, data analytics, and dynamic network analysis. The author also concluded that city disaster management strategies may benefit greatly from the convergence of many research streams by integrating machine learning and crowdsourcing techniques combined with digital twin technology.

To make use of the high-quality datasets generated, the company Avineon is developing an advanced 3D city model with a level of detail LOD2.3. LOD2.3 contain the geometry data of realistic roof modelling for buildings, taking into account dormers and roof overhangs, with an accuracy of 10 cm, and flawless alignment with the 2D large-scale base map, among other things [98,114]. This study was conducted in Jeonju City, South Korea and Houston, Texas. The study collects real-time data on carbon emissions from different sources such as transportation, electricity generation, industrial processes and activities, and commercial and residential. The real-time data collected from those sources are implemented in the city model. The tool used in the study for the city model is the guideline of the Open Geospatial Consortium (OGC) on the level of detail LOD1 and LOD2. In the first stage of the analytics phase, the GIS identified the most susceptible region based on carbon emissions in accordance with IPCC. Next, the radial basis function kernel approach was used to anticipate carbon emission spatial trends. In the last, using a backpropagation neural network, the most influential factor among the data sources is identified (electricity, city gas, domestic garbage, and vehicle). Similarly, Ref. [51] also explains the uses of all LOD 1–4 in the making of the GIS-based city virtual models for flood assessments and their challenges and gaps in the research.

Another popular platform used by various researchers for making digital twin technology is CityGML. As discussed earlier in Section 3.1, CityGML is a conceptual model and exchange format for the representation, storage, and exchange of virtual 3D city models. Until now, it has mostly been employed for modelling buildings, owing to their prominent position within the urban environment and a lack of data for other themed items, such as landscapes, roads, subterranean networks and utilities, water bodies, and other such infrastructure. However, in recent years, it is increasingly being explored for creating digital twins of cities. Additional data is added to the model through various real-time sensors, allowing the model to visualise the city’s high-resolution detail. Similar research has been conducted and explained previously, including [51,63,115,120,121,122,126].

## 4. Discussion

Based on an examination of 68 papers about 3D city modelling, early warning systems, and digital twins, we highlighted that these three categories are interrelated and essential for the creation of digital twin technology for enhancing the climate resilience of smart cities. Many scholars have worked in the past on the 3D modelling of smart cities and the production of various environmental parameter forecasts. Through 3D city modelling, urban designers and stakeholders can work more effectively on approaches to manage the city from different hazards such as floods, storm surges, and sea-level rise. Similarly, such models benefit citizens via many applications such as emergency response, severe event alerts, weather updates, and public awareness on various social media platforms. However, in the works to date relating to 3D city modelling, real-time data has not been used. Conversely, in the field of early warning systems, researchers employed real-time sensor data for the monitoring of various environmental hazards. A digital twin is the combination of both real-time data provided by various sensors and 3D model data to create an accurate digital replica which is linked to the physical city. Meanwhile, the question is raised why we need the digital twin of a city. The past 60 years have seen a massive expansion in population from approximately 3 billion in 1960 to around 7.9 billion in 2021 [2], as well as an increase in infrastructure construction. This makes it very difficult for stakeholders to react swiftly and effectively to climatic event risks. Additionally, cities are impacted by climate change and other natural catastrophes like floods, landslides, and severe temperatures with increasing frequency and severity. City governments around the world have come to accept that their primary goal should be to strengthen their communities’ resilience to natural disasters. As a result, a digital twin paradigm might give significant advantages in terms of improved situation assessment, decision-making, coordination, and resource allocation [124]. Digital twins store data that may be analysed and utilised to make suitable choices for a broad range of activities, from long-term urban planning to emergency response. Digital twins differ from city 3D models in that they are constructed using data that is constantly updated and derived in real-time from a variety of sources [126]. This real-time data is gathered and updated from a variety of sensors via the internet of things (IoT), RFID tags, and smartphones [73]. Since a digital twin utilises real-time, continuously updated data, it is regarded as a more dependable and precise method for forecasting future natural catastrophe occurrences in cities than traditional city models [105,108,111,115].

It is interesting to note that out of 68 evaluated publications, 37 were case studies, 15 were general studies, 5 were review papers, and 11 were conference papers. There were 13 of 37 case studies related to flooding, six to landslide events, and one to each of sea level rise and storm surge. The remainder of the studies were concerned with general environmental issues as shown in Figure 7 and Table 5. These figures highlight that the majority of the studies done in the past on developing 3D smart city-based digital twins are mostly focused on flood events.

According to the Organization for Economic Cooperation and Development (OECD), the incidence of global flooding catastrophes almost quadrupled between 2000 and 2009. Therefore, in response to these risks, researchers are attempting to enhance the flood resilience of their communities via data-driven planning, construction, and operations. Some researchers are doing this via the creation of 3D models and digital twins for smart cities in order to enhance the resilience of existing infrastructure and facilitate continuous development and mitigation measures for these flood threats.

Figure 8 depicts the article outline for developing digital twin technologies for smart cities. It is interesting to note that among the 68 papers, digital twin technology based on a GIS engine seems to provide a new tool for generating an early warning system for extreme climatic hazards.

### 4.1. Challenges and Limitations in the Adoption and Implementation of Digital Twin Technology for Climate Resilience

The lack of actual implementation work in digital twin technology for enhancing climate resilience can be attributed to several reasons. One of the main reasons is the technical challenges in creating and maintaining a digital twin. This includes the requirement for high-quality and high-frequency data from various sources, the difficulty in integrating multiple data sources, and the need for advanced computational methods to process and analyze the data. Furthermore, current hardware and software systems may have limitations, making it challenging to create a representative digital twin.

The lack of understanding and awareness among stakeholders, including policymakers, decision makers, and communities, about the benefits of digital twin technology can limit its adoption and integration into wider environmental management strategies, hindering its ability to enhance climate resilience.

The shortage of funding and resources for research and development has been a hindrance to the advancement of digital twin technology for enhancing climate resilience. However, in recent times, with the availability of funding from the European Union and other sources, researchers are beginning to take an interest in this area. As a result, numerous projects are currently underway (some are outlined in Section 4.2), and it is expected that in a few years, a clear picture of the capabilities of digital twin technology will emerge. With a better understanding of how digital twin technology can be leveraged to increase climate resilience, the future looks promising for the application of this technology in environmental management. It is hoped that this increased focus on digital twin technology will lead to the development of innovative solutions that address the challenges faced by coastal communities.

The adoption and implementation of digital twin technology for climate resilience not only poses technical and financial challenges, it also presents social and ethical challenges. One of the significant concerns is privacy, as digital twins rely on the collection and sharing of large amounts of personal data, raising issues around data privacy and security. Furthermore, the question of data ownership and sharing also arises, as stakeholders from different sectors may have conflicting interests in the data collected by digital twins. This can lead to unequal access to and benefits from technology, creating social and ethical challenges that need to be addressed.

The complexity of the technology and the potential for unintended consequences and errors are also challenges that must be considered. With so many variables involved in creating a digital twin, there is a risk of errors or biases that can affect the outcomes of the model. Such unintended consequences can have significant implications for communities and the environment. Therefore, it is necessary to take measures to ensure that the technology is used responsibly and ethically.

To address these challenges and limitations, effective governance, stakeholder engagement, and collaboration between different sectors and disciplines are essential. These measures will ensure that the technology is used equitably and responsibly, with the interests and needs of all stakeholders considered. In conclusion, the adoption and implementation of digital twin technology for climate resilience should be approached with careful consideration of the social and ethical implications, alongside the technical and financial challenges that need to be addressed.

### 4.2. Future Directions

Due to technological advancement, the concept of the digital twin is of increasing interest in the smart cities sector. Yet, further digital twin research is required in order to develop more effective early warning systems. Numerous projects are now ongoing with the objective of developing digital twin technology. The National Digital Twin program (NDTp) is administered by the Centre for Digital Built Britain, a collaboration between the University of Cambridge and the Department of Business, Energy, and Industrial Strategy. Initiated by HM Treasury in July 2018, their primary objective is to enable a national digital twin—an ecosystem of interconnected digital twins that would deliver an information management framework to guarantee safe, robust data sharing and efficient information management [129].

According to the Geospatial Contribution to Digital Twins for Smart City seminar held by the Hong Kong University of Science and Technology on 19 February 2021, currently, Singapore, Helsinki, Zurich, and Hong Kong are the leading smart cities in the world that are focusing on the development of digital twin technology and its solutions for various environmental issues [130].

Singapore makes use of spatial data to create the smart city [131], Helsinki makes use of spatial data to show suitable living areas for families with children [132], Helsinki also makes use of spatial data to monitor air quality [133], Zurich makes use of spatial data to support recycling in the smart city development [134].

Another example is Lisbon, Portugal where a digital twin drainage master plan has been proposed to deal with the effects of climate change. Flooding has become more common in Lisbon due to rising sea levels and more frequent heavy rainfall occurrences. In the case of a major storm, the current infrastructure is insufficient to provide effective drainage. In the region, 80 flooding events occurred between 1900 and 2006, whereas 15 occurred in the six years between 2008 and 2014. The city of Lisbon has developed a digital twin that allows it to simulate several scenarios and design a mitigation strategy for multiple return periods. When it comes to a drainage master plan for Lisbon, the city’s project team relies on the support of this digital twin model. A novel mitigation approach that prevents 20 floods over the course of 100 years and saves hundreds of millions of euros will be created as a consequence of this technique [135,136]. Another example is a major chemical plant building in Pennsylvania, USA. In order to maintain an up-to-date model of the site, the project team uses drones to fly over it on a weekly basis and analyse the photographs in 2D and 3D. They then use the most recent digital surface model to produce automated 2D flood simulations. Cloud-based web services and a mobile-friendly, web-GIS platform are used to give high-resolution rainfall predictions and flood simulations. For the most part, the goal of this project is to reduce the potential damage that floods may inflict on machines, essential infrastructure, and people, through improving planning and asset management [136].

An article published online by [29], highlights the critical role infrastructure plays in the climate emergency. The article describes an interactive app that explores how digital twins connected to one another can help plan for better adaptation and resilience. For the first time in the UK, CReDo is developing a digital twin across major service networks in order to demonstrate how linked data and increased access to the correct information can help enhance climate adaptation and resilience. Climate change and severe weather, particularly floods, are the focus of this study. It shows how network owners and operators may employ secure, resilient information sharing across sector borders to lessen the impact of floods on network performance and customer service [29].

The European Union (EU) also aims to become carbon neutral by 2050, and the European Green Deal, a collection of measures meant to facilitate this green transition, was launched in December 2019 [137]. Digital transformation into an open and accessible European data space as a major node for well-informed decision making is a vital part of the Green Deal. The project GreenData4All is working to build a European strategy for finding, managing, and using geospatial data. Destination Earth (DestinE) and GreenData4All are two of the most significant EU initiatives aimed at bolstering essential information systems. DestinE is working to create digital twins of the Earth that can be used to track and anticipate environmental change and human influence in support of long-term sustainability. The research team is expecting that, by 2030, the project will be able to construct a complete digital representation of the Earth. Their goals are to create Earth system simulations, and high-resolution predictive models, which will be extremely realistic, interactive, and dynamic. These models will be informed by extensive observational information, such as regional effects of climate change, natural disasters, marine ecosystems, or urban environments.

The other main objective of the DestinE project is the implementation of nature-based solutions. Starting in 2021, the first high-priority digital twins for extremes prediction and climate change adaptation are scheduled to be produced, in accordance with the new Digital Europe financing program. Other EU initiatives, such as the project ‘Smart control of the climate resilience in European coastal cities (SCORE)’, are also aiming to develop digital twin technology to build climate-resilient smart cities. SCORE is a EUR 10m Horizon 2020-funded research project aimed at securing coastal and low-lying areas from increasing climate and sea level risks such as coastal flooding and erosion. The project’s purpose is to use open, accessible digital twin technologies to incorporate citizen scientists in developing prototype coastal city early warning systems and enable smart, real-time climate resilience monitoring and control in European coastal cities [138].

### 4.3. Summary

This systematic literature review highlights that the development of digital twin technology for climate event management is still in an emerging phase of experimental research. Therefore, an increase in the level of maturity of the various elements is needed to realise the concept of digital twins for the management of extreme climate events. In addition, due to the rapidly increasing interest of the European Union and other funding agencies in digital twin technology, the number of publications focusing on climate-resilient smart cities can be expected to increase dramatically in the near future. Consequently, probably, certain enablers and obstacles have not yet been discovered, and the categories of constituent technologies outlined here may need further expansion in future.

There is currently intense interest from researchers in digital twin concepts for enhancing the climate resilience of cities. However, as in the case of any new intervention, their usefulness must be proven before a majority of policy makers and stakeholders in the vulnerable cities can support their adoption. Moreover, to get the support of local communities, real-world examples of digital twins with proven benefits and real results are needed. Currently, there are only theoretical discussions about the role of digital twins in building climate-resilient smart cities.

## 5. Conclusions

In the context of climate-resilient smart cities, this article provides a comprehensive analysis of concepts such as digital twin technology, 3D city modelling, and early warning systems employing real-time sensor data. This study investigates the connections between the early warning system and the 3D city modelling ideas, bringing them together to generate a new technology for climate-resilience enhancement termed a ‘Digital twin’. As a systematic review, this research drew on previous case studies and analyses of digital twin technologies, as well as contributory methodologies and their framework, to provide insights into both experimental observations and theoretical literature. The main focus of this study is on the extreme climate context, specifically addressing challenges such as coastal flooding, erosion, storm surges, and sea level rise. This systematic review highlights the potential of 3D city modelling, early warning systems, and digital twins in the creation of technology for enhancing the climate resilience of smart cities. While numerous studies have proposed concepts of digital twin technology, including generating early warning alerts for flood events using digital twins, data management and governance issues must be addressed to fully realize its potential. It is important to note that the digital twin, in the context of environmental management, is still in the emerging phase of experimental research. However, with the increased interest in digital twin technology from the European Union and other public bodies, the number of publications focusing on climate-resilient smart cities is expected to grow significantly in the near future.

From an academic perspective, this systematic review contributes to the body of literature on digital twin technology by exploring its enabling technologies and potential application in the context of smart cities. Furthermore, it highlights the importance of addressing data management and governance issues to maximize the benefits of digital twin technology. From a managerial perspective, the findings of this study could be used to guide the development and implementation of climate-resilient digital twin technology in smart cities. However, it is essential to acknowledge the limitations of this review, such as the exclusion of studies not written in English or not available in full-text format.

For future research, it is recommended to conduct case studies that explore the practical application of digital twin technology in real-world settings. Additionally, further research should be conducted on the economic and social implications of implementing digital twin technology in smart cities. Finally, there is a need for more in-depth research on the data management and governance issues associated with digital twin technology. In conclusion, this is a pioneer systematic review to examine the potential of the concept of digital twin technology employed in the context of smart cities, and in particular, climate resilience.

## Figures and Tables

**Figure 1 sensors-23-02659-f001:**
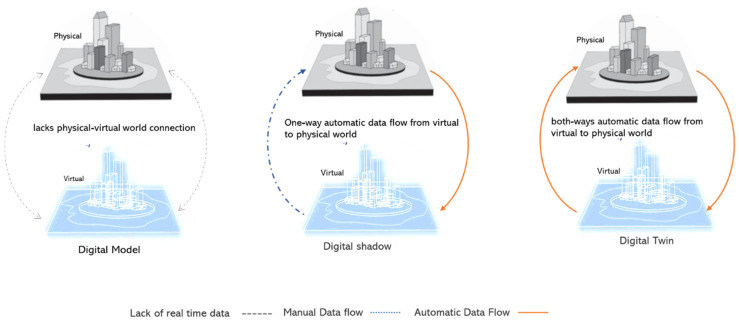
Illustration of the difference between digital model (**left**), Digital shadows (**centre**), and digital twin (**right**) with the aid of automatic and manual data flow between the virtual and real world.

**Figure 2 sensors-23-02659-f002:**
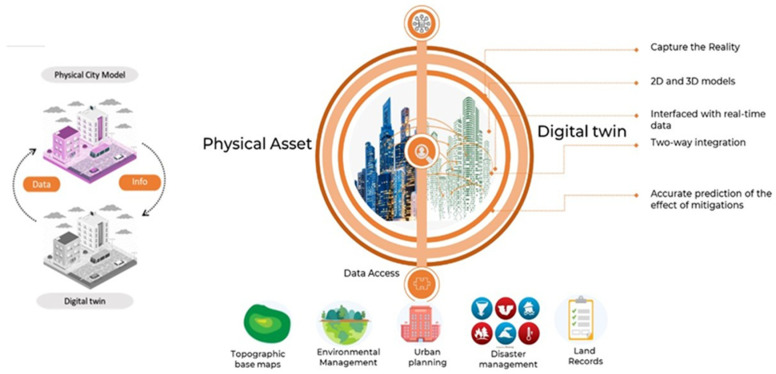
Sensor data allows physical city assets to be transferred to a digital twin model, allowing the city and its subsystems to be modified.

**Figure 3 sensors-23-02659-f003:**
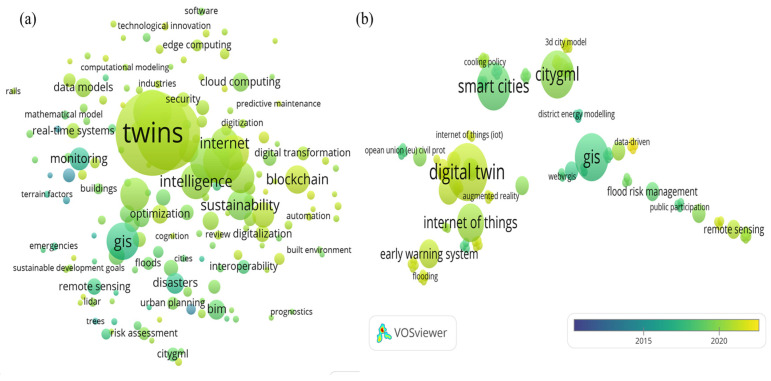
Keyword density visualisation with a time frame (**a**) first screening (**b**) second screening.

**Figure 4 sensors-23-02659-f004:**
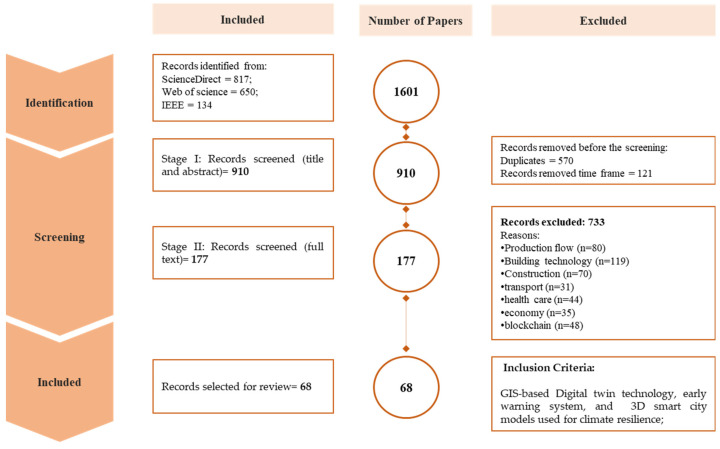
Schematic representation for the selection of literature reviewed based on the PRISMA approach.

**Figure 5 sensors-23-02659-f005:**
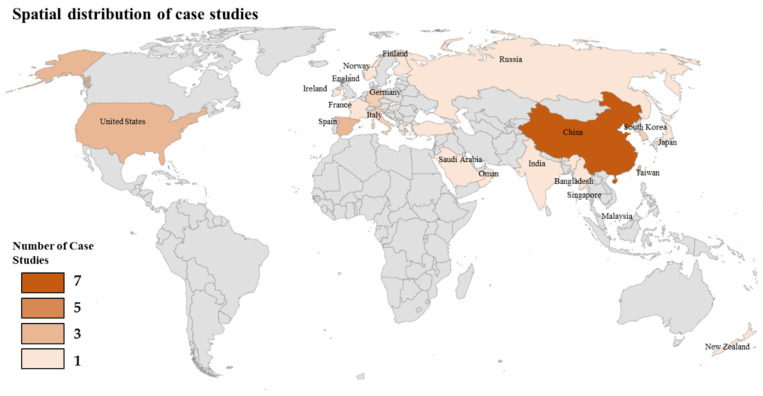
Spatial distribution of past case studies conducted on digital twins for management of extreme climate events in smart cities.

**Figure 6 sensors-23-02659-f006:**
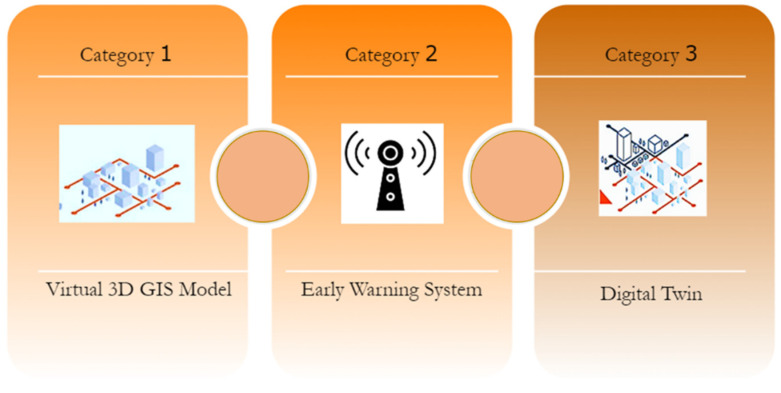
Key themes Identified for Study: A visual overview.

**Figure 7 sensors-23-02659-f007:**
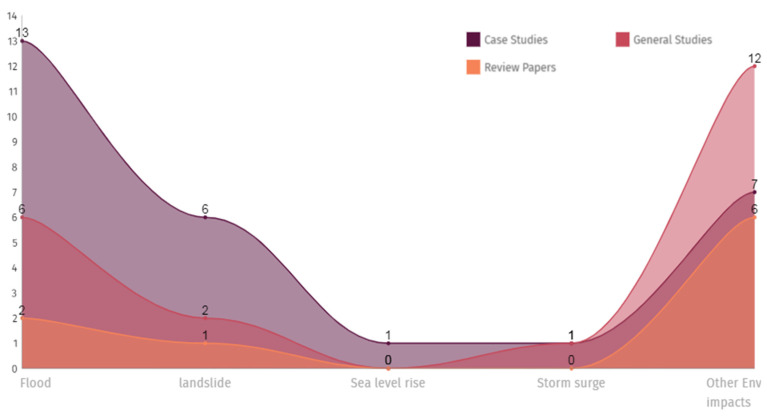
Number of studies reviewed according to different environmental parameters.

**Figure 8 sensors-23-02659-f008:**
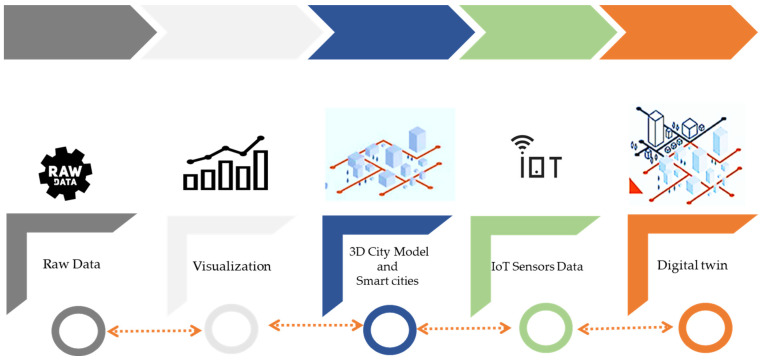
Step-by-step representation of article outline for the development of a digital twin technology for smart cities.

**Table 1 sensors-23-02659-t001:** List of search queries conducted in this study.

#	Search String: 1	Search String: 2	Search String: 3
	ScienceDirect (n = 817)	Web of Science (n = 650)	IEEE (n = 134)
1	“Digital Twin” AND “GIS-Based” (n = 18)	“Digital Twin” AND “GIS-Based” (n = 18)	“Digital Twin” AND “GIS-Based” (n = 8)
2	“Digital Twin” AND “Climate Resilience” (n = 8)	“Digital Twin” AND “Climate Resilience” (n = 3)	“Digital Twin” AND “Climate Resilience” (n = 1)
3	Digital Twin” AND “Smart City” (n = 344)	Digital Twin” AND “Smart City” (n = 95)	Digital Twin” AND “Smart City” (n = 74)
4	Digital Twin” AND “Climate change” (n = 261)	Digital Twin” AND “Climate change” (n = 29)	Digital Twin” AND “Climate change” (n = 6)
5	“Digital Twin” AND “City Model” AND “Climate “(n = 112)	“Digital Twin” AND “City Model” AND “Climate “(n = 482)	“Digital Twin” AND “City Model” AND “Climate “(n = 24)
6	“Digital Twin” AND “Early warning system” AND “Sensor “(n = 74)	“Digital Twin” AND “Early warning system” AND “Sensor “(n = 32)	“Digital Twin” AND “Early warning system” AND “Sensor “(n = 21)

**Table 5 sensors-23-02659-t005:** Distribution of reviewed studies based on various environmental parameters.

Categories	Parameters	Citations
Digital Twin	Floods	[94,97,107,108,125]
	Landslide	[105]
	Sea Level Rise/Storm Surge	
	Building Environment	[37,105,109,110,112,113,114,115,116,123,127]
	Smart Cities	[37,93,94,98,102,107,117,118,119,120,121,122]
Virtual 3-D GIS Model	Floods	[44,45,46,47,55,56,57,58,61,62,128]
	Landslide	None
	Sea Level Rise/Storm Surge	[45,55]
	Building Environment	[53,63,64]
	Smart Cities	
Early Warning System	Floods	[69,71,73,74,77,88,90,91]
	Landslide	[17,68,70,72,75,78,79,82,83,86,89]
	Sea Level Rise/Storm Surge	[80]
	Building Environment	[86]
	Smart Cities	[87]

## Data Availability

Not applicable.

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
