# Peer review of "Management of Climate Resilience: Exploring the Potential of Digital Twin Technology, 3D City Modelling, and Early Warning Systems"

_sensors, 2023, doi:10.3390/s23052659_

Round 1
Reviewer 1 Report
This paper attempts to survey the area of digital twins as regards the management of extreme climate events in smart cities. On a first glance, the paper seems promising and is of interest to the readers of this journal, however, there are a number of significant flaws in the current version of the text.
First of all, the title is a bit misleading, since digital twins are only a third of the papers considered here - from the title, I would expect to read mainly about the use of digital twins for the management of extreme climate events in cities, which is not the case for this particular paper. So, I would suggest to rethink the title of the paper and instead use something closer to the actual contents of the paper.
Second, the way the related work was chosen and, thus, presented, does not really cover the way digital twins are currently being used to manage extreme climate events. I would consider this as a significant flow. I would thus suggest to the authors to change the title, and also adapt the abstract, as well as work on making the 3 areas presented a bit less disjoint between them. For sure, it could also be a theme in the discussion section to discuss if, how, and why these areas are currently the way they are, and what could be done in the future to remedy this situation.
There is also some degree of inconsistency in how certain terms are referenced, e.g., I am not sure the authors are referring to Geodesign (https://en.wikipedia.org/wiki/Geodesign) or whether "Geo Design" is something different. There are similar issues throughout the text, so I would urge the authors to carefully check (and also add references to the definitions of) such terms.
I am not sure about the dates the papers included were published, the authors mention a range between 2022 and 2021 in line 198, which is probably a mistake. If the December 2021 is the final cut-off date, I think it would be a good idea to at least include some months from 2022, since the field of digital twins in smart cities seems to be evolving quickly in the last couple of years. E.g. a relevant recent paper from 2022 is the following:
A. Sulasikin, Y. Nugrahat, M. E. Aminanto, B. I. Nasution and J. I. Kanggrawan, "Developing a knowledge management system for supporting flood decision-making," 2022 IEEE International Smart Cities Conference (ISC2), Pafos, Cyprus, 2022, pp. 1-4, doi: 10.1109/ISC255366.2022.9921881.
Other than the above comments, the introduction needs some minor rewrites in places, in order to make the text more coherent - right now, paragraphs are a bit disconnected.
As regards the presentation of the paper, I think the authors need to revisit their text and revise/correct certain parts. E.g. some mistakes:
- Lots of sentences are missing a full stop "." at the end.
- line 44: "and?"
- a "." is missing at the end of line 61
- "According to the Smart City Index (SCI), Singapore, Zurich, and Oslo are the top three nations in the development of smart cities." you probably mean *cities* and not *nations*
- "Other single case studies are conducted over France, Ireland, UK, Germany, India Russia, Japan, Saudi Arabia, Oman, Finland, Malaysia and New Zealand.": Greece seems to be missing here from what is shown in Figure 5.
- "With rising population and urbanisation, it is becoming more challenging to manage the impact on cities from different environmental catastrophic events [48—50]. As cities are increasing in size and complexity it is becoming more difficult to predict and manage the impact of climate change.": the second sentence is probably redundant.
Overall, the paper is interesting, but has some flaws, which can be corrected if the text is revised. I would propose that the authors conduct a major revision of their paper and resubmit.
Reviewer 2 Report
This paper may need to make some improvements.
1. The introduction is too long. It includes the literature review. Please provide a new section of the literature review. Also, I cannot see clearly the research setting or gaps in the introduction section.
2. In the conclusion section, please provide the academic contribution, managerial implications, limitations, and future research direction.
Round 2
Reviewer 1 Report
This is a revision of an interesting paper attempting to present the current situation as regards the climate resilience through the use of digital twin, city modelling and early warning systems. The paper essentially attempts to survey the application of such fields in the current context of climate resilience.
Overall, I find the changes made by the authors to the original manuscript to have significantly improved the quality the text, and to have addressed many of its flaws. I also think that the paper is more balanced now, and that the new title reflects much better the actual content of the paper.
I think that the newly added Challenges subsection could be further enhanced, in order to provide some better understanding of the area presented in the paper.
Some additional minor comments:
- Maybe the authors could revisit Figure 2 and add some descriptions to either the figure (better) or the caption, e.g., in the case of the digital model "no automatic or manual flow between the physical and the virtual world" etc.
- There are still some minor spellchecking issues with the text, please make a pass through the text to correct them.
- Some sentences are still not very well connected to the rest of the text, e.g., in line 456 "One of the benefits of ultrasonic detection is its exceptional capacity to non-destructively penetrate into an item. Sulasikin et al..." probably the first could be omitted entirely.
- A number of sentences begin with a reference, e.g., lines 619, 624, 633, etc., which could be better connected to the rest of the text - in terms of style it is also not the best option to have consecutive paragraphs beginning with a reference, so I would recommend to edit such parts in the text.
Overall, I recommend to accept the paper as is, since it has been greatly enhanced in this revision, but would also recommend to the authors to check the above minor comments and also try to expand the Challenges section a bit.
